# Potential of Urine Biomarkers CHI3L1, NGAL, TIMP-2, IGFBP7, and Combinations as Complementary Diagnostic Tools for Acute Kidney Injury after Pediatric Cardiac Surgery: A Prospective Cohort Study

**DOI:** 10.3390/diagnostics13061047

**Published:** 2023-03-09

**Authors:** Wim Vandenberghe, Jorien De Loor, Katrien Francois, Kristof Vandekerckhove, Ingrid Herck, Johan Vande Walle, Harlinde Peperstraete, Thierry Bové, Daniël De Wolf, Lieve Nuytinck, Jan J. De Waele, Evelyne Meyer, Eric A. J. Hoste

**Affiliations:** 1Department of Intensive Care Medicine, Ghent University Hospital, Ghent University, 9000 Ghent, Belgium; 2Department of Cardiac Surgery, Ghent University Hospital, Ghent University, 9000 Ghent, Belgium; 3Department of Pediatric Cardiology, Ghent University Hospital, Ghent University, 9000 Ghent, Belgium; 4Department of Internal Medicine and Pediatrics, Faculty of Medicine and Health Sciences, Ghent University, 9000 Ghent, Belgium; 5Department of Pediatric Nephrology, ERKNET Centre, Ghent University Hospital, Ghent University, 9000 Ghent, Belgium; 6Health, Innovation and Research Institute UZ Gent, Ghent University Hospital, 9000 Ghent, Belgium; 7Laboratory of Biochemistry, Department of Veterinary and Biosciences, Faculty of Veterinary Medicine, Ghent University, 9000 Ghent, Belgium; 8Research Foundation-Flanders (FWO), 1000 Brussels, Belgium

**Keywords:** AKI, urine biomarker, pediatric, cardiac surgery

## Abstract

Acute kidney injury (AKI) is common after pediatric cardiac surgery (CS). Several urine biomarkers have been validated to detect AKI earlier. The objective of this study was to evaluate urine CHI3L1, NGAL, TIMP-2, IGFBP7, and NephroCheck^®^ as predictors for AKI ≥ 1 in pediatric CS after 48 h and AKI ≥ 2 after 12 h. Pediatric patients (age < 18 year; body weight ≥ 2 kg) requiring CS were prospectively included. Urine CHI3L1, NGAL, TIMP-2, IGFBP7, and NephroCheck^®^ were measured during surgery and intensive care unit (ICU) stay and corrected for urine dilution. One hundred and one pediatric patients were included. AKI ≥ 1 within 48 h after ICU admission occurred in 62.4% and AKI ≥ 2 within 12 h in 30.7%. All damage biomarkers predicted AKI ≥ 1 within 48 h after ICU admission, when corrected for urine dilution: CHI3L1 (AUC-ROC: 0.642 (95% CI, 0.535–0.741)), NGAL (0.765 (0.664–0.848)), TIMP-2 (0.778 (0.662–0.868)), IGFBP7 (0.796 (0.682–0.883)), NephroCheck^®^ (0.734 (0.614–0.832)). Similarly, AKI ≥ 2 within 12 h was predicted by all damage biomarkers when corrected for urine dilution: uCHI3L1 (AUC-ROC: 0.686 (95% CI, 0.580–0.780)), NGAL (0.714 (0.609–0.804)), TIMP-2 (0.830 (0.722–0.909)), IGFBP7 (0.834 (0.725–0.912)), NephroCheck^®^ (0.774 (0.658–0.865)). After pediatric cardiac surgery, the damage biomarkers urine CHI3L1, NGAL, TIMP-2, IGFBP7, and NephroCheck^®^ reliably predict AKI after correction for urine dilution.

## 1. Introduction

Acute kidney injury (AKI) is a common complication after pediatric cardiac surgery (CS), with a reported incidence varying between 28% and 86% [1,2,3,4,5].

According the Kidney Disease: Improving global Outcomes (KDIGO) definition, AKI is diagnosed based on serum creatinine level (sCr) and urine output (UO) [6,7]. Urine biomarkers, reflecting damage to the kidney, have been validated in various clinical settings [8,9,10,11]. Damage biomarkers allow earlier detection of AKI, recognition of different AKI phenotypes, and evaluation and quantification of the effect of an intervention [8,9,10,11]. The ultimate goal of using biomarkers is to detect patients at high risk for AKI, so measures can be taken to prevent AKI.

In 2005, Mishra et al. evaluated urine neutrophil gelatinase-associated lipocalin (NGAL) as a novel biomarker for AKI after pediatric CS [12].

The two urine cell cycle arrest biomarkers, which are tissue inhibitorof metalloproteinases-2 (TIMP-2) and insulin-like growth factor-binding protein 7 (IGFBP7), measured by the point-of-care NephroCheck^®^ test, had better test characteristics than NGAL for early diagnosis of moderate to severe AKI (KDIGO stage 2 or 3) within 12 h of sample collection in adult critically ill patients [13]. Data on pediatric CS patients are limited for this test.

Another biomarker, urine chitinase 3-like protein 1 (CHI3L1), has previously been evaluated as a predictive tool for AKI in the adult ICU setting [14,15].

In this study, we aimed to evaluate CHI3L1 as a damage AKI biomarker and compare it with other damage biomarkers (NGAL, TIMP-2, IGFBP7, and NephroCheck^®^) in pediatric patients after CS.

## 2. The Materials and Methods

This single-center cohort study was conducted at the Ghent University Hospital during a 1.5-year study period.

The study is reported according to the STrengthening the Reporting of OBservational studies in Epidemiology (STROBE) Statement (Appendix A) [16].

### 2.1. Study Patients

Pediatric patients (age < 18 years with body weight ≥ 2 kg) who underwent elective CS were enrolled preoperatively. Exclusion criteria were AKI stage ≥ 1 on admission, stage 5 chronic kidney disease (CKD), kidney transplantation, lack of informed consent, and surgery during the weekend (Figure 1).

### 2.2. Definitions

#### 2.2.1. Acute Kidney Injury

AKI was diagnosed and classified according to the KDIGO definition using both SCr and UO criteria [3]. KDIGO added one sCr criterion to the definition of AKI stage 3 for use in children: ‘In patients < 18 years (y), decrease in estimated GFR (eGFR) to <35 mL/min per 1.73 m^2^’. eGFR was calculated with the original Schwartz estimate equation (Appendix A) [18,19,20]. Since sCr does not have stable levels in neonates ≤ 7 days old, the eGFR criterion was not used in these patients [21]. Baseline sCr was defined as the lowest value within the last 3 months prior to enrollment, including preop serum creatinine. Two investigators (J.D.L. and E.A.J.H.) determined the clinically relevant baseline sCr. Urine output was hourly registered in the patient management system and validated by ICU nurses.

#### 2.2.2. Subclinical AKI_NGAL_

Urine NGAL was measured at ICU admission and 4 h after admission. Bennett et al. measured NGAL in a healthy pediatric population ≥ 3 years old and reported that the 95th percentile value (57.6 ng/mL) could serve as a guide for the upper limit of the normal range [22]. As proposed by Bennett et al., we determined in children less than 3 years old without AKI the 95th percentile value of NGAL at ICU admission (28.3 ng/mL). 

#### 2.2.3. Subclinical AKI_NephroCheck_^®^


The 0.3 (ng/mL)^2^/1000 cutoff of the NephroCheck^®^ test identifies patients with high risk for AKI [23]. In our pediatric cohort, subclinical AKI_NephroCheck_^®^ was defined as not meeting AKI criteria and a NephroCheck^®^ value ≥ 0.3 (ng/mL)^2^/1000 4 h after ICU admission.

### 2.3. Primary and Secondary Endpoints

The primary endpoint was the occurrence of AKI ≥ 1 within 48 h after ICU admission; the secondary endpoint was the occurrence of AKI ≥ 2 within 12 h after ICU admission.

Investigated AKI biomarkers included urine CHI3L1, NGAL, TIMP-2, IGFBP7, NephroCheck^®^, and the difference between the postoperative and preoperative functional biomarker sCr (ΔsCr(postop-preop)). Differences in biomarker concentration between patients with and without AKI were investigated, as well as the diagnostic performance of biomarkers and combinations thereof.

### 2.4. Ethics Approval and Consent to Participate

The Ethics Committee of Ghent University Hospital approved this study (B670201213147). In patients < 12 years old, both parents provided written informed consent (IC) preoperatively, while in patients ≥ 12 years old, the patient and both parents provided written IC. The study was performed according to the declaration of Helsinki and Good Clinical Practice Guidelines.

### 2.5. Trial Workflow

Urine and blood samples were taken between induction of anesthesia and the start of CS at ICU admission and 2, 4, 6, 12, 24, 48 h after ICU admission (Figure 2).

Sampling was performed in the operating room and in the ICU only. Blood samples were collected according to the European Directive on blood volume limits for sampling [24]. Serum and urine supernatants were stored at −80 °C and thawed at room temperature immediately prior to analysis.

Clinical data were extracted from the hospital records. The maximum inotropic score (ISmax) and vasoactive-inotropic score (VISmax) were calculated as described by Wernovsky et al. and Gaies et al. [25,26].

Individual CS procedures were classified in different categories according to the ‘Risk Adjusted Classification for Congenital Heart Surgery’ (RACHS-1) [27]. Samples were anonymized, as were clinical data. All technicians were blinded to clinical data.

### 2.6. Biomarker Analysis

Urine biomarkers were analyzed in urine samples, collected from the urinary catheter, which is routinely placed in patients undergoing cardiac surgery. Urine CHI3L1, TIMP-2, IGFBP7, and NephroCheck^®^ analyses were performed at the Laboratory of Biochemistry, Ghent University Hospital. The concentration of CHI3L1 was measured by a human sandwich enzyme-linked immunosorbent assay (ELISA) (DC3L10, R&D Systems, Minneapolis, MN, USA) validated for urine, whereas the Astute140^®^ Meter measured the concentrations of urine TIMP-2, urine IGFBP7, and NephroCheck^®^ by a fluorescent immunoassay (NephroCheck^®^ Test, Astute Medical, San Diego, CA, USA). The Cobas c502 autoanalyzer measured the concentration of creatine by a kinetic rate blanked Jaffé assay (Roche Diagnostics, Basel, Switzerland). Urine NGAL analysis was performed at the central laboratory of Sint-Lucas Bruges Hospital. The Siemens Dimension Vista measured the concentration of NGAL by a particle-enhanced turbidimetric immunoassay (ST001-3CA, BioPorto, Hellerup, Denmark). Details on these laboratory analyses were previously described by our group [14]. In addition, the standard sample dilution scheme used in CHI3L1 ELISA (supplementary Appendix A) and the adjustment of NGAL (Appendix A), TIMP-2, and IGFBP7 concentrations (Appendix A) before input in statistical programs are provided.

Based on the temporal relationship of the predictive value of the biomarker, NGAL was measured at ICU admission and 4 h later, whereas CHI3L1, TIMP-2, IGFBP7, and NephroCheck^®^ were measured only 4 h after ICU admission [28].

### 2.7. Biomarker Diagnostic Test Adjustments

Fluid administration and use of diuretics might influence urine biomarker concentration [29]. Similar to, e.g., albuminuria, the investigated urine biomarkers were normalized by dividing the urine biomarker concentration by the urine creatinine (uCr) concentration [30]. Ralib et al. showed that normalizing for uCr improved the performance of urine biomarkers for AKI [29].

Biomarkers were also evaluated as a combination in two- or three-biomarker panels. Additionally, the absolute difference between post- and preoperative sCr (ΔsCr[postop-preop]) was evaluated as a diagnostic test.

### 2.8. Statistical Analysis

Descriptive statistics and the unpaired comparison of variables between two independent samples was performed using SPSS 25 (IBM, Armonk, NY, USA). Categorical variables were analyzed using Fisher’s exact or the chi-square test and continuous variables using the nonparametric Mann–Whitney U test. Additionally, the 95% CI was calculated for a proportion using the Wilson procedure without a correction for continuity [31,32]. For all analyses, two-sided *p*-values < 0.050 were considered statistically significant.

The predictive performance of biomarkers for AKI was assessed by calculating the area under the receiver operating characteristics curve (AUC-ROC) using MedCalc 15.2.1 (MedCalc^®^ Software, Ostend, Belgium). A biomarker or panel was considered to have failed as a reliable test to predict AKI when the 95% CI of the AUC-ROC contained the value 0.500. AUC-ROC results were compared by using the method by Delong et al. in MedCalc^®^ 15.2.1.

## 3. Results

### 3.1. Patients

Of the 106 enrolled pediatric patients, 5 were excluded due to missing reference sCr (*n* = 2) and the presence of AKI stage ≥ 1 at enrollment (*n* = 3) (Figure 1). Of the 101 included pediatric patients, 54.5% were male, 50% were <1 year old, and 10% were premature. In 65% of the patients, this was the first CS procedure; cardiopulmonary bypass pump (CPB) was used in 91%; and in 97% of the patients, RACHS-1 was equal to or below 3 (Table 1).

### 3.2. Acute Kidney Injury

#### 3.2.1. Occurrence of AKI

Serum creatinine was available for all patients until 48 h after ICU admission. AKI ≥ 1 occurred in 63 out of 101 patients (62.4%) within 48 h after ICU admission of which 29 (28.7%) were classified as stage 1, 18 (17.8%) as stage 2, and 16 (15.8%) as stage 3. The diagnosis of AKI was mainly based on sCr criteria (Table 2). AKI was present in 4 (4%) patients at ICU admission. Those 4 patients had AKI stage 1 based on creatinine criteria, and all evolved to AKI stage 2 during ICU stay. In the first week after CS, 3 patients (3.0%) were treated with kidney replacement therapy (KRT).

Compared with patients without AKI, patients with AKI were younger and had lower preoperative and reference sCr, less comorbidities, higher postoperative hemoglobin, higher postoperative ISmax and VISmax scores, higher RACHS-1 score, more CPB use, lower priming volume of the CPB pump, and longer duration of CS. Additional perioperative characteristics are provided in Appendix A. AKI patients were not different in weight or length, corrected for gender and age, compared with patients without AKI (Appendix A). AKI patients had a longer median ICU length of stay (5 versus 3 days; *p* = 0.002) and hospital length of stay (9 days versus 6; *p* ≤ 0.001); ICU mortality was 2.0% (Table 1).

#### 3.2.2. Occurrence of Subclinical AKI

For 89 out of 101 patients (88.0%), a urine sample was available 4 h after ICU admission. NephroCheck^®^ test results were available for 76 patients (85%) due to a limited number of tests available. In 71 out of 101 patients, both NGAL and NephroCheck^®^ were measured. Subclinical AKI_NGAL_ occurred in 2 out of 71 patients (2.8%), while subclinical AKI_NephroCheck_^®^ occurred in 6 out of 71 patients (8.5%). When using both NGAL and NephroCheck^®^, subclinical AKI was present in 8 out of 71 patients (11.3%). 

### 3.3. Primary Endpoint: AKI ≥ 1 within 48 h

There was no difference in urine biomarker concentrations measured 4 h after ICU admission between patients with and without AKI ≥ 1 (Table 3). All biomarkers failed as a reliable diagnostic test for AKI ≥ 1 within 48 h after ICU admission (Table 4a and Figure 3).

When biomarker concentrations were normalized for urine dilution, the AUC-ROC of CHI3L1, NGAL, TIMP-2, IGFBP7, and NephroCheck^®^ was significant for predicting AKI ≥ 1 within 48 h after ICU admission (Table 4a and Figure 3). The performance of CHI3L1 as an acute kidney damage biomarker was inferior compared with NGAL and IGFBP7 (resp *p* = 0.036; *p* = 0.008), but similar to the other investigated biomarkers (Table 4a). Four hours after ICU admission, the AUC-ROC of the functional AKI biomarker ΔsCr[postop-preop] was 0.911 (95% CI 0.806–0.970) for AKI ≥ 1. The combination of biomarkers, corrected for urine dilution, did not result in higher AUC-ROC values compared with the separate biomarkers (Table 4b and Figure 3).

### 3.4. Secondary Endpoint: AKI ≥ 2 within 12 h

Biomarker concentrations measured 4 h after ICU admission were not different between patients with and without AKI ≥ 2 (Table 3). All biomarkers failed as diagnostic tests for AKI ≥ 2 within 12 h after ICU admission (Table 4a and Figure 3).

When normalized for urine dilution, all biomarkers had a significant association with the occurrence of AKI ≥ 2 within 12 h after ICU admission (Table 4a and Figure 3). CHI3L1 performed significantly inferiorly compared with TIMP-2 and IGFBP7 (resp *p* = 0.028; *p* = 0.005), but similar to the other investigated acute kidney damage biomarkers (Table 4a). The AUC-ROC of the functional AKI biomarker ΔsCr[postop-preop] was 0.886 (95% CI 0.775–0.955). The combination of biomarkers, corrected for urine dilution, did not result in higher AUC-ROC values compared with the separate biomarkers (Table 4b and Figure 3).

### 3.5. Sensitivity Analysis

A sensitivity analysis was performed to evaluate biomarker performance to predict AKI ≥ 1 in patients who did not already have AKI ≥ 1 at ICU admission, and a similar analysis was performed for AKI ≥ 2. The results are presented in Appendix A When corrected for urine dilution, all biomarkers predicted AKI. The combination of biomarkers did not result in higher AUC-ROC values (Appendix A).

## 4. Discussion

In this single-center prospective cohort study on pediatric CS patients, all tested acute kidney damage biomarkers predicted the occurrence of AKI ≥ 1 within 48 h and AKI ≥ 2 within 12 h after ICU admission, when their concentration was corrected for urine dilution.

The goal of detecting patients at risk for AKI is to implement measures to prevent AKI. KDIGO proposed a bundle of AKI preventive measures such as close monitoring of renal function, optimization of hemodynamics and intravascular volume status, and avoidance of hyperglycemia, nephrotoxic drugs, and contrast agents. Two small studies in adult patients after cardiac surgery showed that the implementation of such a bundle in biomarker-positive patients indeed reduced the incidence of AKI [28]. Whether this approach prevents AKI in a pediatric cardiac surgery cohort should be investigated.

Absolute biomarker concentrations in our study were markedly lower compared with concentrations in previous pediatric studies. Median NGAL 4 h after ICU admission was 0.1 ng/mL (*IQR* 0.1–20.0), and median NephroCheck^®^ was 0.18 (ng/mL)^2^/1000 (*IQR* 0.07–0.5) in AKI ≥ 1 patients. Urine NGAL levels reported at the same time by Dong et al. and Meers et al. were >1000-fold higher, i.e., 307 ng/mL (*IQR* 225–418) and 200 ng/mL (SE 50), respectively [33,34]. For NephroCheck^®^, about 5- to 10-fold higher values were also reported by Dong et al., Gist et al., and Meersch et al., i.e., 0.96 (ng/mL)^2^/1000 (*95% CI* 0.61–1.50); 0.8 (ng/mL)^2^/1000 (*IQR* 0.46–1.83), and 2.0 (ng/mL)^2^/1000(SE 0.4) [33,34,35]. There are several potential explanations for the lower biomarker concentrations in our study. First, the hit on the kidneys may have been less profound in our cohort. In our cohort, almost 10% of the patients underwent CS without CPB use. Since CPB use is associated with AKI, this may explain the lower occurrence of AKI compared with other cohorts. Second, it has been reported that biomarker concentrations are different in different age groups [36]. Bojat et al. revealed a significant association between the basal biomarker concentration and age at the time of surgery, with lower values in younger patients [36]. Westhoff et al. measured stable biomarker values among age groups in children up to 9 years old admitted on ICU without AKI, which tended to be lower in neonates and younger children [37]. Patients in our study had similar ages compared with Gist et al., i.e., respectively, 6 months (IQR 4–31) versus 5 months (SD 3), but were far younger compared with those in the other studies [33,34,35]. A third possible explanation for the observed differences is that urine dilution occurred more often in our cohort compared with others. In contrast, Dong et al., Meersch et al., and Tao et al. reported significantly higher biomarker concentrations in AKI after pediatric CS already without correction for dilution [33,35,38]. Urine concentration is the result of administered fluid volume and/or diuretic use or ultrafiltration use on CPB. In our center, anesthetists routinely administer diuretics at the end of the procedure. Therefore, concentration may explain the high occurrence of AKI based on serum creatinine. On the other hand, increased urine output as a consequence of exposure to diuretics may have diluted the urine biomarker concentration. Another explanation for the high AKI occurrence based on SCr is the low baseline values of SCr with a lower quartile of 0.2. This means that a creatinine rise to 0.3 is already enough to achieve AKI stage 1.

It should be acknowledged that the occurrence of AKI itself may induce a bias when biomarkers are corrected for urine dilution. Indeed, in AKI, urine creatinine concentration will be lower, which has an impact on the correction for dilution. Likewise, Cr production is decreased in smaller children, potentially leading to higher biomarker corrections, when corrected for ‘dilution’. Nonetheless, both NGAL and NephroCheck^®^, when corrected for urine dilution, were able to predict AKI in our pediatric cohort with AUC-ROC values > 0.700. Previous studies corroborate the AKI predicting capability of NGAL in children after CS. Peco-Antić et al. showed NGAL AUC-ROC of 0.700; Meersch et al., 0.850; and even AUC-ROC values above 0.900 were presented by Krawczeskiet et al.(0.910), Dong et al. (0.913), and Galić et al. (0.930) [33,34,39,40,41]. Previous studies also support the predictive performance of NephroCheck^®^ for AKI in children after CS, albeit not for stage 1. Tao et al., Meersch et al., and Dong et al. calculated AUC-ROC values for AKI ≥ 2 of 0.730, 0.850, and 0.733, respectively. In contrast, in patients <1 year who underwent CS, NephroCheck^®^ lacked AKI predicting capability in a study by Bojan et al. The impact of age on biomarker concentration, the very early timing (i.e., 3 h after CPB) of sampling, and urine dilution are potential explanations for these neutral results [36].

Urine CHI3L1 was associated with AKI, but with lower AUC-ROC values compared with other biomarkers studied. This is in contrast to our previous findings in adult ICU patients, but in line with those in adults after CS [15,42]. Our findings may indicate another underlying pathophysiology of AKI in CS and general ICU patients. Potentially, the CHI3L1 signaling pathway is more activated in a septic versus in a sterile inflammation context associated with CS. This suggested hypothesis requires further investigation.

The ΔsCr[postop-preop] seems to be an excellent functional AKI predictor, but using sCr as an AKI predictor is obviously a self-fulfilling prophecy, because it is used both as a predictor and as an outcome variable.

### Strengths and Limitations

This study has several strengths. First, it is the first study evaluating urine CHI3L1 in a pediatric population. Second, several (combinations of) biomarkers were evaluated prospectively in more than 100 pediatric patients. Third, both sCr and UO were used to define AKI as intended by KDIGO compared with most studies only using sCr with potential underestimation of AKI occurrence. Fourth, this study evaluated early diagnosis by biomarkers, which allows early identification of patients at risk for AKI and subsequent targeted use of a therapy bundle [28].

Nevertheless, there are several limitations as well. First, the single-center design with protocols specific to our center may not be generalizable to other centers. Second, NephroCheck^®^ was not evaluated in all patients. Third, although investigated by others in a pediatric population, NephroCheck^®^ is intended to be used in patients 21 years of age or older. Fourth, similar to other studies in this domain, this study includes a heterogeneous cohort of pediatric CS patients. Finally, damage biomarkers detect damage, while serum creatinine identifies reduced kidney function. The difference between biomarkers can be used to define phenotypes of AKI [43]. A patient may have subclinical AKI, defined as damage detected by a damage biomarker, while normal serum creatinine indicates absence of functional AKI. Additionally, patients with functional AKI defined as a rise in serum creatinine may have absence of signs of damage and so a negative reading of a damage biomarker.

## 5. Conclusions

After pediatric cardiac surgery, damage AKI biomarkers urine CHI3L1, NGAL, TIMP-2, IGFBP7, and NephroCheck^®^ reliably predict AKI, but only after correction for urine dilution.

## Figures and Tables

**Figure 1 diagnostics-13-01047-f001:**
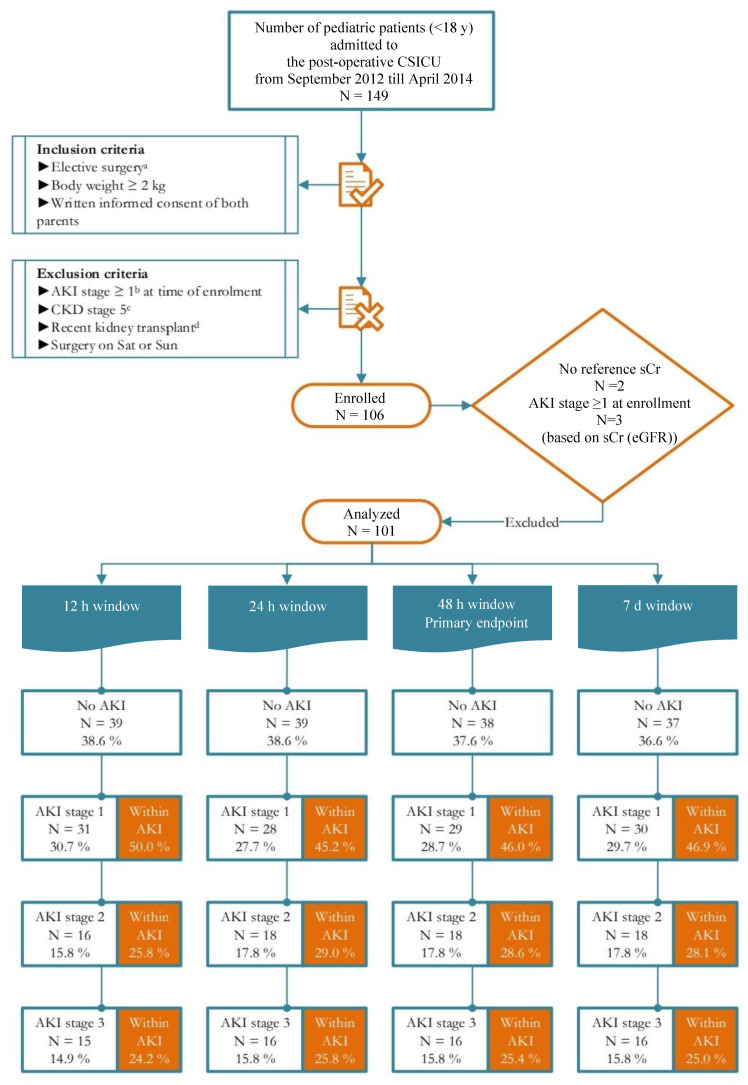
Study flow diagram. ^a^ Planned ≥ 4 h in advance; ^b^ KDIGO definitions for the diagnosis and staging of AKI, which are based on sCr and UO [3]; ^c^ KDOQI definitions for the diagnosis and staging of CKD [17]; ^d^ ≤3 mo before; AKI: acute kidney injury; CKD: chronic kidney disease; CSICU: cardiac surgery intensive care unit; h: hour; eGFR: estimated glomerular filtration rate; KDIGO: Kidney Disease: Improving Global Outcomes; KDOQI: Kidney Disease Outcomes Quality Initiative; mo: month; No.: number; Sat: Saturday; sCr: serum creatinine; Sun: Sunday; UO: urine output; y: year.

**Figure 2 diagnostics-13-01047-f002:**
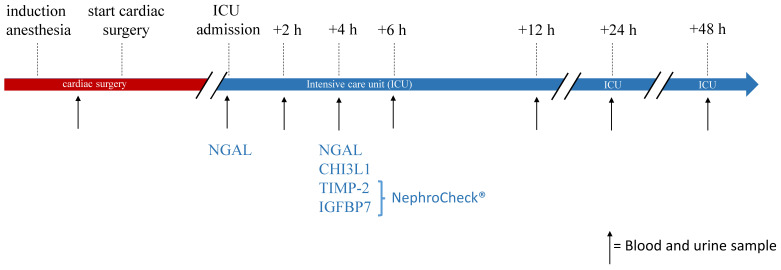
Trail workflow: timing of blood and urine sampling. ICU: intensive care unit; NGAL: neutrophil gelatinase-associated lipocalin; TIMP-2: tissue inhibitor of metalloproteinases-2; IGFBP7: insulin-like growth factor-binding protein 7.

**Figure 3 diagnostics-13-01047-f003:**
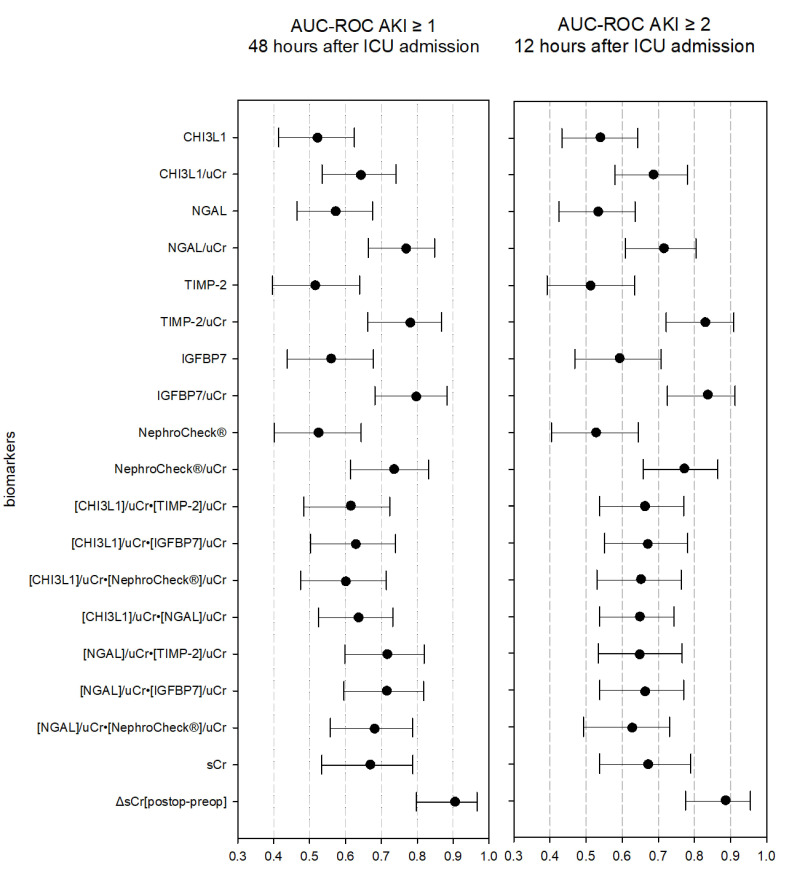
Performance of biomarkers and combinations of biomarkers in predicting acute kidney injury with and without correction for urine dilution. AKI: acute kidney injury; AUC-ROC: area under the receiver-operating characteristics curve; IGFBP7: insulin-like growth factor-binding protein 7; NephroCheck^®^: two-biomarker panel urine [TIMP-2]•[IGFBP7]; sCr: serum creatinine; TIMP-2: tissue inhibitor of metalloproteinases-2; CHI3L1: chitinase 3-like protein 1; uCr: urine creatinine; NGAL: neutrophil gelatinase-associated lipocalin.

**Table 1 diagnostics-13-01047-t001:** Characteristics of the pediatric patients, the cardiac surgery procedures, and outcomes.

	All Patients (*n* = 101)	AKI Stage ≥ 1 within 48 h (*n* = 63)	No AKI within 48 h (*n* = 38)	*p*-Value
PATIENT DEMOGRAPHIC CHARACTERISTICS				
Male sex—no. (%) (95% CI)	55 (54.5)	35 (55.6)	20 (52.6)	0.838
Caucasian—no. (%) (95% CI)	98 (97.0)	60 (95.2)	38 (100)	0.289
Age (IQR)—y	0 (0–4)	0 (0–2)	3 (1–11)	<0.001
Age groups				<0.001
<6 w (<42 d)	10 (9.9)	8 (12.7)	2 (5.3)	
≥6 w and <1 y	41 (40.6)	34 (54.0)	7 (18.4)	
≥1 y and <10 y	37 (36.6)	18 (28.6)	19 (50.0)	
≥10 y and <18 y	13 (12.9)	3 (4.8)	10 (26.3)	
PATIENT HEALTH CHARACTERISTICS				
Medical history				
Preoperative sCr (IQR)—mg/dL (*n* = 100)	0.3 (0.2–0.4)	0.3 (0.2–0.4)	0.4 (0.3–0.6)	<0.001
Reference sCr (IQR)—mg/dL	0.3 (0.2–0.4)	0.3 (0.2–0.4)	0.4 (0.3–0.6)	<0.001
Comorbidities—no. (%) (95% CI)				0.038
Asthma	2 (2.0) (0.5–6.9)	0 (0.0) (0.0–5.7)	2 (5.3) (1.5–17.3)	
Down syndrome	11 (10.9) (6.2–18.5)	8 (12.7) (6.6–23.1)	3 (7.9) (2.7–20.8)	
Dysmorphism	2 (2.0) (0.5–6.9)	0 (0.0) (0.0–5.7)	2 (5.3) (1.5–17.3)	
RSV infection in medical history	2 (2.0) (0.5–6.9)	1 (1.6) (0.3–8.5)	1 (2.6) (0.5–13.5)	
Other	13 (12.9) (7.7–20.8)	5 (7.9) (3.4–17.3)	8 (21.1) (11.1–36.3)	
None	71 (70.3) (60.8–78.3)	49 (77.8) (66.1–86.3)	22 (57.9) (42.2–72.1)	
Premature at birth (gestational age at birth <38 w)	10 (9.9) (5.5–17.3)	8 (12.7) (6.6–23.1)	2 (5.3) (1.5–17.3)	0.312
Number of previous non-CS surgeries—no. (%) (95% CI)				0.145
1	7 (6.9) (3.4–13.6)	2 (3.2) (0.9–10.9)	5 (13.2) (5.8–27.3)	
2	2 (2.0) (0.5–6.9)	1 (1.6) (0.3–8.5)	1 (2.6) (0.5–13.5)	
None	92 (91.1) (83.9–95.2)	60 (95.2) (86.9–98.4)	32 (84.2) (69.6–92.6)	
Number of previous CS—no. (%) (95% CI)				0.314
1	21 (20.8) (14.0–29.7)	11 (17.5) (10.0–28.6)	10 (26.3) (15.0–42.0)	
2	10 (9.9) (5.5–17.3)	7 (11.1) (5.5–21.2)	3 (7.9) (2.7–20.8)	
3	4 (4.0) (1.6–9.7)	4 (6.3) (2.5–15.2)	0 (0.0) (0.0–9.2)	
None	66 (65.3) (55.7–73.9)	41 (65.1) (52.8–75.7)	25 (65.8) (49.9–78.8)	
Preoperative clinical examination				
Blood pressure (IQR)—mm Hg				
Systolic	96 (86–107)	92 (81–103)	101 (92–114)	0.004
Diastolic	58 (48–67)	55 (47–67)	60 (50–67)	0.430
Mean	71 (61–80)	69 (59–79)	74 (65–84)	0.076
Heart rhythm—no. (%) (95% CI) (*n* = 73)				NA
Normal sinus rhythm	73 (100) (95.0–100)	44 (100) (92.0–100)	29 (100) (88.3–100)	
Heart rate in normal sinus rhythm (IQR)—bpm (*n* = 73)	119 (98–135)	126 (105–134)	110 (92–136)	0.215
Left ventricle ejection fraction—no. (%) (95% CI) (*n* = 95)				0.286
≤20%	0 (0.0) (0.0–3.9)	0 (0.0) (0.0–6.1)	0 (0.0) (0.0–9.6)	
21–30%	0 (0.0) (0.0–3.9)	0 (0.0) (0.0–6.1)	0 (0.0) (0.0–9.6)	
31–50%	3 (3.2) (1.1–8.9)	3 (5.1) (1.7–13.9)	0 (0.0) (0.0–9.6)	
>50%	92 (96.8) (91.1–98.9)	56 (94.9) (86.1–98.3)	36 (100) (90.4–100)	
Fractional shortening (*n* = 72)	38 (34–42)	39 (34–42)	38 (34–42)	0.916
Preoperative medication—no. (%) (95% CI)				
ACE inhibitors	2 (2.0) (0.5–6.9)	2 (3.2) (0.9–10.9)	0 (0.0) (0.0–9.2)	0.526
Diuretics	4 (4.0) (1.6–9.7)	4 (6.3) (2.5–15.2)	0 (0.0) (0.0–9.2)	0.294
NSAIDs	1 (1.0) (0.2–5.4)	0 (0.0) (0.0–5.7)	1 (2.6) (0.5–13.5)	0.376
Corticosteroids	3 (3.0) (1.0–8.4)	0 (0.0) (0.0–5.7)	3 (7.9) (2.7–20.8)	0.051
Iodinated contrast <72 h before CS	1 (1.0) (0.2–5.4)	1 (1.6) (0.3–8.5)	0 (0.0) (0.0–9.2)	1.000
Postoperative clinical examination				
Left ventricle ejection fraction—no. (%) (*n* = 91)				NA
>50%	91 (90.1)	57 (90.5)	34 (89.4)	
Fractional shortening (*n* = 80)	38 (35–42)	37 (35–41)	38 (36–43)	0.270
Hemoglobin (IQR)—g/dL	11.6 (10.3–12.6)	12.2 (10.9–12.9)	10.7 (9.8–12.1)	0.002
Wernovsky ISmax at dCS	0 (0–2)	0 (0–4)	0 (0–0)	0.001
VISmax ad dCS	2 (0–6)	4 (0–10)	0 (0–3)	0.001
Wernovsky ISmax at dPO1	0 (0–0)	0 (0–0)	0 (0–0)	0.010
VISmax at dPO1	0 (0–4)	0 (0–6)	0 (0–0)	0.001
Highest VISmax	2 (0–7)	4 (0–13)	0 (0–3)	0.001
CHARACTERISTICS OF THE CS PROCEDURE				
RACHS-1 method—no. (%) (95% CI)				0.044
Risk category 1	14 (13.9) (8.4–21.9)	4 (6.3) (2.5–15.2)	10 (26.3) (15.0–42.0)	
Risk category 2	46 (45.5) (36.2–55.2)	30 (47.6) (35.8–59.7)	16 (42.1) (27.9–57.8)	
Risk category 3	37 (36.6) (27.9–46.4)	25 (39.7) (28.5–52.0)	12 (31.6) (19.1–47.5)	
Risk category 4	3 (3.0) (1.0–8.4)	3 (4.8) (1.6–13.1)	0 (0.0) (0.0–9.2)	
Risk category 5	0 (0.0) (0.0–3.7)	0 (0.0) (0.0–5.7)	0 (0.0) (0.0–9.2)	
Risk category 6	1 (1.0) (0.2–5.4)	1 (1.6) (0.3–8.5)	0 (0.0) (0.0–9.2)	
CPB—no. (%) (95% CI)	91 (90.1) (82.7–94.5)	60 (95.2) (86.9–98.4)	31 (81.6) (66.6–90.8)	0.038
Duration of CPB (IQR)—min (*n* = 91)	71 (55–98)	78 (58–98)	65 (50–98)	0.158
Priming volume of CPB pump (IQR)—ml (*n* = 79)	180 (170–350)	170 (170–270)	300 (175–800)	< 0.001
Duration of aortic clamp during CPB (IQR)—min (*n* = 83)	44 (22–62)	49 (20–64)	39 (24–62)	0.384
Duration of surgery (IQR)—h (*n* = 100)	4.1 (3.5–4.8)	4.3 (3.7–5.1)	4.0 (3.4–4.4)	0.035
SHORT-TERM OUTCOMES				
KRT in period dCS-dPO7—no. (%) (95% CI)	3 (3.0) (1.0–8.4)	3 (4.8) (1.6–13.1)	0 (0.0) (0.0–9.2)	0.289
CSICU mortality—no. (%) (95% CI)	2 (2.0) (0.5–6.9)	2 (3.2) (0.9–10.9)	0 (0.0) (0.0–9.2)	0.526
CSICU LOS (IQR)—d	3 (2–5)	3 (2–6)	3 (2–3)	0.026
Total ICU LOS (IQR)—d	4 (3–7)	5 (3–8)	3 (3–4)	0.002
Hospital LOS (IQR)—d	8 (6–13)	9 (7–15)	6 (6–8)	<0.001

ACE: angiotensin-converting enzyme; AKI: acute kidney injury; BMI: body mass index; bpm: beats per minute; CI: confidence interval; CS: cardiac surgery: CSICU: cardiac surgery intensive care unit; d: day; dCS: day of cardiac surgery; dPO1: postoperative day 1; CPB: cardio pulmonary bypass; h: hour; IQR: interquartile range; ISmax: maximum inotrope score; KDIGO: Kidney Disease: Improving Global Outcomes; LOS: length of stay; no: number; NSAID: nonsteroidal anti-inflammatory drug; RACHS-1: Risk Adjusted Classification for Congenital Heart Surgery; KRT: kidney replacement therapy; RSV: respiratory syncytial virus; sCr: serum creatinine; UO: urine output; VISmax: maximum vasoactive-inotropic score; w: week, y: year.

**Table 2 diagnostics-13-01047-t002:** Acute kidney injury based on KDIGO criteria.

	12 h after ICU Admission			24 h after ICU Admission			48 h after ICU Admission		
	sCr Only *n* (%)	UO Only *n* (%)	sCr and UO *n* (%)	sCr Only *n* (%)	UO Only *n* (%)	sCr and UO *n* (%)	sCr Only *n* (%)	UO Only *n* (%)	sCr and UO *n* (%)
No AKI	39 (38.6)	100 (99.0)	39 (38.6)	39 (38.6)	95 (94.1)	39 (38.6)	38 (37.6)	93 (92.1)	38 (37.6)
AKI 1	31 (30.7)	1 (1.0)	31 (30.7)	28 (27.7)	4 (4.0)	28 (27.7)	29 (28.7)	5 (5.0)	29 (28.7)
AKI 2	16 (15.8)	0 (0.0)	16 (15.8)	18 (17.8)	2 (2.0)	18 (17.8)	18 (17.8)	2 (2.0)	18 (17.8)
AKI 3	15 (14.9)	0 (0.0)	15 (14.9)	16 (15.8)	0 (0.0)	16 (15.8)	16 (15.8)	1 (1.0)	16 (15.8)

ICU: intensive care unit; AKI: acute kidney injury; KDIGO: Kidney Disease: Improving Global Outcomes; sCr: serum creatinine; UO: urine output.

**Table 3 diagnostics-13-01047-t003:** Acute kidney stress biomarkers in pediatric patients with and without acute kidney injury.

**Primary outcome**	** *n* **	**No AKI**	**AKI ≥ 1**	***p*-Value**
CHI3L1	89	0.18 (0.02–0.48)	0.16 (0.02–0.37)	0.745
NGAL	89	0.10 (0.10–10.05)	0.10 (0.10–20.00)	0.153
TIMP-2	76	4.30 (2.63–6.45)	3.60 (2.70–5.80)	0.786
IGFBP7	76	35.30 (21.70–64.88)	40.40 (21.80–100.20)	0.383
NephroCheck^®^	76	0.15 (0.08–0.28)	0.17 (0.07–0.54)	0.729
**Secondary outcome**		**No AKI**	**AKI ≥ 2**	***p*-Value**
CHI3L1	89	0.18 (0.02–0.48)	0.16 (0.02–0.37)	0.745
NGAL	89	0.10 (0.10–10.05)	0.10 (0.10–20.00)	0.153
TIMP-2	76	4.30 (2.63–6.45)	3.60 (2.70–5.80)	0.786
IGFBP7	76	35.30 (21.70–64.88)	40.40 (21.80–100.20)	0.383
NephroCheck^®^	76	0.15 (0.08–0.28)	0.17 (0.07–0.54)	0.729

AKI: acute kidney injury; NGAL: neutrophil gelatinase-associated lipocalin; TIMP-2: tissue inhibitor of metalloproteinases-2; IGFBP7: insulin-like growth factor-binding protein 7; CHIL3L1, NGAL; TIMP-2; IGFBP7 are expressed as ng/mL; NephroCheck^®^ as (ng/mL)^2^/1000. All biomarkers were measured in urine. Data are presented as median and 25th and 75th interquartile ranges.

**Table 4 diagnostics-13-01047-t004:** (a) Predictive value of acute kidney injury by biomarkers measured 4 h after ICU admission in urine. (b) Predictive value of acute kidney injury by biomarker combinations measured 4 h after ICU admission in urine.

**(a)**					
**Outcome**	** *n* **	**AKI ≥ 1 within 48 h after ICU Admission** **AUC-ROC (95% CI)**	**AKI ≥ 2 within 12 h after ICU Admission** **AUC-ROC (95% CI)**	***p*-Value** **Compared with CHI3L1/uCr** **for** **AKI ≥ 1 within 48 h after ICU Admission**	***p*-Value** **Compared with CHI3L1/uCr** **for** **AKI ≥ 2 within 12 h after ICU Admission**
CHI3L1	89	0.520 (0.414–0.625)	0.539 (0.433–0.643)		
CHI3L1/uCr	89	0.642 (0.535–0.741)	0.686 (0.580–0.780)		
NGAL	89	0.572 (0.465–0.675)	0.532 (0.425–0.636)		
NGAL/uCr	89	0.765 (0.664–0.848)	0.714 (0.609–0.804)	0.036	0.674
NephroCheck^®^	76	0.524 (0.402–0.644)	0.526 (0.404–0.645)		
NephroCheck^®^/uCr	76	0.734 (0.614–0.832)	0.774 (0.658–0.865)	0.230	0.101
TIMP-2	76	0.519 (0.397–0.639)	0.514 (0.392–0.635)		
TIMP-2/uCr	76	0.778 (0.662–0.868)	0.830 (0.722–0.909)	0.117	0.028
IGFBP7	76	0.560 (0.438–0.678)	0.593 (0.469–0.708)		
IGFBP7/uCr	76	0.796 (0.682–0.883)	0.834 (0.725–0.912)	0.008	0.005
sCr	101	0.670 (0.534–0.788)	0.673 (0.537–0.790)		
ΔsCr[postop-preop]	101	0.911 (0.806–0.970)	0.886 (0.775–0.955)	0.001	0.142
**(b)**					
**Outcome**	** *n* **	**AKI ≥ 1 within 48 h after ICU Admission** **AUC-ROC (95% CI)**	**AKI ≥ 2 within 12 h after ICU Admission** **AUC-ROC (95% CI)**		
**Combinations**					
**All Corrected for uCr**					
[CHI3L1]•[TIMP-2]	76	0.608 (0.484–0.723)	0.661 (0.538–0.770)		
[CHI3L1]•[IGFBP7]	76	0.627 (0.503- 0.740)	0.674 (0.551–0.781)		
[CHI3L1]•[NephroCheck^®^]	76	0.599 (0.475–0.714)	0.654 (0.530- 0.763)		
[CHI3L1]•[NGAL]	89	0.633 (0.525–0.732)	0.645 (0.537–0.743)		
[NGAL]•[TIMP-2]	76	0.718 (0.598–0.819)	0.657 (0.534–0.766)		
[NGAL]•[IGFBP7]	76	0.716 (0.596–0.818)	0.661 (0.538–0.770)		
[NGAL]•[NephroCheck^®^]	76	0.681 (0.558–0.787)	0.617 (0.493–0.731)		

Data are presented as AUC-ROC values and 95% confidence interval. sCr: serum creatinine; CHI3L1: chitinase 3-like protein 1; uCr: urine creatinine; NGAL: neutrophil gelatinase-associated lipocalin; TIMP-2: tissue inhibitor of metalloproteinases-2; IGFBP7: insulin-like growth factor-binding protein 7; NephroCheck^®^: AKI risk by analyzing TIMP-2and IGFBP7. Biomarkers/uCr: correction of the biomarker for urine dilution. *p*-Values are calculated to compare AUC-ROC by the Delong et al. method.

## Data Availability

The datasets generated during and/or analyzed during the current study are available from the corresponding author on reasonable request.

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
