# Peer review of "Potential of Urine Biomarkers CHI3L1, NGAL, TIMP-2, IGFBP7, and Combinations as Complementary Diagnostic Tools for Acute Kidney Injury after Pediatric Cardiac Surgery: A Prospective Cohort Study"

_diagnostics, 2023, doi:10.3390/diagnostics13061047_

Round 1

Reviewer 1 Report

The article by Vandenberghe et al. described the possibility of using biomarkers from urine to detect AKI after Pediatric Cardiac Surgery. The cohort has included satisfying number of population for each condition and exclusion conditions were properly implemented. The article has some spelling errors and editing issues which must be corrected before publication.

1.       If the mentioned biomarkers are from urine why author say only uNGAL and not for other markers?

2.       Please be consistent while using the word. Either use either “pediatrics” or “paediatrics”.

3.       Instead of using “ damage biomarkers” please use “Injury biomarkers”

4.       Page 2 line 46 please remove damage before the kidney injury.

5.       Page 2 line 61 remove “2” and “materials and methods”

6.       Page 3 line 93 remove space between  AKINephroCheck® and was defined.

7.       Line 105 please remove damage before the AKI biomarkers. “Investigated AKI biomarkers” is good. The same mistake was observed in other places in the manuscript. Editing of the entire manuscript for extra spaces ,repetitive words is necessary .

8.       Figure 2. trail please Capitalize “t”

Reviewer 2 Report

Authors analyzed urine biomarkers as a predictive tool of AKI after cardiac surgery in pediatric patients. The work was well performed and had little to argue with, since AKI studies were always complicated by too many confounding factors. As a nephrology practitioner, readers will definitely be eager to know how to apply the results in real world. Will authors propose any more aggressive treatment strategies to prevent the development of AKI? This may add more value to this paper.
